# WWOX Loses the Ability to Regulate Oncogenic AP-2γ and Synergizes with Tumor Suppressor AP-2α in High-Grade Bladder Cancer

**DOI:** 10.3390/cancers13122957

**Published:** 2021-06-12

**Authors:** Damian Kołat, Żaneta Kałuzińska, Andrzej K. Bednarek, Elżbieta Płuciennik

**Affiliations:** Department of Molecular Carcinogenesis, Medical University of Lodz, 90-752 Lodz, Poland; zaneta.kaluzinska@stud.umed.lodz.pl (Ż.K.); andrzej.bednarek@umed.lodz.pl (A.K.B.); elzbieta.pluciennik@umed.lodz.pl (E.P.)

**Keywords:** *WWOX*, *TFAP2A*, *TFAP2C*, AP-2alpha, AP-2gamma, AP-2 transcription factors, bladder cancer

## Abstract

**Simple Summary:**

A prognostic factor of bladder cancer which can benefit from research on molecular markers is the histologic grade. WWOX, AP-2α and AP-2γ are known to have a grade-dependent effect but have not been investigated in that aspect in bladder cancer. Depending on the specific collaboration, their role was found to promote or inhibit grade 2 bladder cancer. As further research is needed on higher grades, the aim of the present study was to examine the functionality of WWOX and two AP-2 factors in grade 3 and grade 4 bladder cancer. It was found that WWOX, AP-2α and their combination mainly demonstrated anti-cancer properties; in contrast, AP-2γ promoted cancer development, which was not inhibited by WWOX in their combined variant. Next-generation sequencing was performed to identify the genes worth investigating as biomarkers. To conclude, WWOX and AP-2α demonstrate tumor suppressor synergism in high-grade bladder cancer, similar to intermediate grade. However, WWOX does not appear to guide oncogenic AP-2γ, which was the case in the lower grade. The cause of such a change in molecule superiority, as well as proposed bladder cancer-related genes, should be further investigated.

**Abstract:**

The cytogenic locus of the *WWOX* gene overlaps with the second most active fragile site, FRA16D, which is present at a higher frequency in bladder cancer (BLCA) patients with smoking habit, a known risk factor of this tumor. Recently, we demonstrated the relevance of the role of WWOX in grade 2 BLCA in collaboration with two AP-2 transcription factors whose molecular actions supported or opposed pro-cancerous events, suggesting a distinct character. As further research is needed on higher grades, the aim of the present study was to examine WWOX-AP-2 functionality in grade 3 and 4 BLCA using equivalent in vitro methodology with additional transcriptome profiling of cellular variants. WWOX and AP-2α demonstrated similar anti-cancer functionality in most biological processes with subtle differences in MMP-2/9 regulation; this contradicted that of AP-2γ, whose actions potentiated cancer progression. Simultaneous overexpression of WWOX and AP-2α/AP-2γ revealed that single discrepancies appear in WWOX-AP-2α collaboration but only at the highest BLCA grade; WWOX-AP-2α collaboration was considered anti-cancer. However, WWOX only appeared to have residual activity against oncogenic AP-2γ in grade 3 and 4: variants with either AP-2γ overexpression alone or combined WWOX and AP-2γ overexpression demonstrated similar pro-tumoral behavior. Transcriptome profiling with further gene ontology certified biological processes investigated in vitro and indicated groups of genes consisting of AP-2 targets and molecules worth investigation as biomarkers. In conclusion, tumor suppressor synergism between WWOX and AP-2α is unimpaired in high-grade BLCA compared to intermediate grade, yet the ability of WWOX to guide oncogenic AP-2γ is almost completely lost.

## 1. Introduction

Chromosomal instability at regions known as Common Fragile Sites (CFSs) is an important aspect of carcinogenesis, since the induction of changes in these areas entails further loss of tumor suppressor genes or amplification of oncogenes [1,2]. Three tumor suppressor genes residing on the most active sites of this type are Fragile Histidine Triad Diadenosine Triphosphatase (*FHIT*) at FRA3B, WW Domain Containing Oxidoreductase (*WWOX*) at FRA16D and Parkinson Disease-2 (*PARK2*) at FRA6E [3,4].

The second most fragile gene, *WWOX*, became the subject of our research, since in BLCA patients, the frequency of chromosome breakage at FRA16D increases when patients have a smoking habit [5], which is a risk factor for that cancer [6]. Moreover, allelic loss at *WWOX*-proximal locus 16q24 was found to be related to tumor progression in up to 45% of BLCAs [7]. Recently, we examined the role of WWOX in conjunction with two interacting proteins, which together orchestrated biological processes diversely in BLCA depending on their cellular level [8]. These were two Activating Enhancer-Binding Protein 2 (AP-2) Transcription Factors (TFs), AP-2α and AP-2γ, members of the AP-2 family, whose natural function is related to early development regulation [9], but which are also involved in carcinogenesis [10,11,12,13,14], including BLCA [15,16,17]. AP-2α and AP-2γ have distinct roles in different tumors, and tend to play an anti-cancer or tumor-promoting role, respectively [9].

Our previous study found that the interaction between WWOX and AP-2α or AP-2γ has different mechanisms of action in BLCA. It was found that both WWOX and AP-2α hinder cancer progression and may act synergistically. In contrast, AP-2γ supports tumor development, but its action may be diminished by WWOX, resulting in a net anti-tumor effect [8]. Nonetheless, the conclusion of previous work was based on grade 2 (G2) bladder carcinoma; hence, profound examination is advisable on higher grades of BLCA. Tumor grading reflects the degree of dedifferentiation (anaplasia) and is an important factor that predicts biological aggressiveness of bladder cancer, affecting patient outcome [18,19,20]; it can also benefit from molecular marker research [21]. The link between tumor grading and WWOX/AP-2α/AP-2γ appears significant, while only the former protein is documented in BLCA [22]. The difference between two AP-2 factors indicates that the loss of AP-2α is associated with increasing glioma grade [23], while AP-2γ overexpression is linked to a higher breast cancer grade [24]. This emphasizes the need for research on the relationship between tumor grade and these molecules in bladder cancer. Therefore, the aim of the present study was to investigate the functionality of WWOX, AP-2α and AP-2γ in G3 and G4 bladder cancer and identify changes in biological processes between bladder cancer grades. Equivalent in vitro cellular variants were used to remain consistent with the preceding study; additionally, transcriptome profiling, gene ontology, protein–protein network visualization and assessment of biomarker utility have been performed.

## 2. Materials and Methods

### 2.1. Cell Lines and Culture Conditions

The urinary bladder carcinoma cell lines HT-1376 and CAL-29 were purchased from Deutsche Sammlung von Mikroorganismen und Zellkulturen (DSMZ, Brunswick, Germany). HT-1376 cells were cultured in high-glucose (4.5 g/L) DMEM medium supplemented with 15% heat-inactivated FBS (Thermo Fisher Scientific, Naarden, the Netherlands), while CAL-29 cells in DMEM medium were supplemented with 10% heat-inactivated FBS (Thermo Fisher Scientific, Naarden, the Netherlands); for both cell lines, the media were also complemented with 1% L-glutamine (Thermo Fisher Scientific, Naarden, the Netherlands) and 1% Antibiotic-Antimycotic (100 units/mL of penicillin, 100 µg/mL of streptomycin and 0.25 µg/mL of Fungizone; Thermo Fisher Scientific, Naarden, the Netherlands). Cells were incubated at 37 °C in a humidified atmosphere of 5% CO_2_.

### 2.2. Stable Transductions

The GIPZ™ lentiviral system (pLenti-GIII-CMV-GFP-2A-Puro) was chosen to overexpress WWOX, with Puro-Blank Lentivirus as control (Applied Biological Materials Inc., Richmond, BC, Canada). The cells were transduced in starvation medium with 8 µg/mL polybrene (Merck Life Sciences Sigma Aldrich, Darmstadt, Germany) and lentiviral particles at MOI = 3. After 24 h of incubation, the viral medium was changed to full medium. Antibiotic-based clone selection was performed using 1 µg/mL puromycin (Merck Life Sciences Sigma Aldrich, Darmstadt, Germany) after 72 h.

The aforementioned cellular variants, i.e., with WWOX overexpression or without (control), were exposed to a second stable transduction. The Lentiviral system pLKO-Neo-CMV and its control Puro-Blank Lentivirus (Merck Life Sciences Sigma Aldrich, Darmstadt, Germany) were used to perform AP-2α or AP-2γ overexpression. Variants were transduced in starvation medium containing 8 µg/mL polybrene (Merck Life Sciences Sigma Aldrich, Darmstadt, Germany) and lentiviral particles at MOI = 3. Changing the starvation medium to the full medium after 24 h was followed by clone selection using 400 μg/mL G418 (Merck Life Sciences Sigma Aldrich, Darmstadt, Germany) for 3 weeks.

For both HT-1376 and CAL-29 cell lines, stable transductions allowed to develop the following variants (abbreviations in brackets):Variants with an appropriate level of WWOX and AP-2α expression:
○Control/control (K/K); Control/AP-2α↑ (K/A); WWOX↑/control (W/K); WWOX↑/AP-2α↑ (W/A)
Variants with an appropriate level of WWOX and AP-2γ expression:
○Control/control (K/K); Control/AP-2γ↑ (K/C); WWOX↑/control (W/K); WWOX↑/AP-2γ↑ (W/C)

Specific variants were compared to the appropriate controls established through the first stable transduction (Figure 1). Cellular variants with only AP-2α or AP-2γ overexpression (K/A or K/C, respectively) were compared to K/K, the same in terms of WWOX overexpression alone (W/K). Variants with overexpression of both WWOX and AP-2α or AP-2γ (W/A or W/C, respectively) were compared to W/K.

### 2.3. Protein Extraction and Western Blot

The RIPA Lysis Buffer supplemented with sodium orthovanadate, phosphatase inhibitor cocktail and phenylmethylsulfonyl fluoride (Santa Cruz Biotechnology Inc., Dallas, TX, USA) was used for cell lysis and protein isolation. Protein concentration was determined using the Bradford method (Bio-Rad Laboratories, Hercules, CA, USA), after which 10% SDS-PAGE and wet transfer to a PVDF membrane (Merck Life Sciences Sigma Aldrich, Darmstadt, Germany) were performed. Staining with Ponceau red (Merck Life Sciences Sigma Aldrich, Darmstadt, Germany) confirmed proper transfer and was followed by membrane blocking using 5% non-fat milk in 1X TBST buffer (Merck Life Sciences Sigma Aldrich, Darmstadt, Germany) for 1 h at room temperature. Anti-WWOX (PA5-29701, Thermo Fisher Scientific, Naarden, the Netherlands) or anti-AP-2α/AP-2γ (respectively GTX113564 or GTX134259, GeneTex Inc., Irvine, CA, USA) were added as primary antibodies diluted 1:1000 with 1% non-fat milk in 1X TBST solution. After overnight incubation, membranes were washed three times using 1X TBST buffer and then incubated with goat anti-rabbit secondary antibodies conjugated with alkaline phosphatase (Merck Life Sciences Sigma Aldrich, Darmstadt, Germany). The intensity of the bands was visualized with Novex^®^ AP Chromogenic Substrate (Invitrogen Life Technologies, Carlsbad, CA, USA) and estimated using ImageJ densitometric analysis software [25] with anti-GAPDH as reference (sc-59540, Santa Cruz Biotechnology Inc., Dallas, TX, USA). The assay was performed in triplicate and duplicate for the first and second stable transduction, respectively.

### 2.4. Immunocytochemistry

The ice-cold ethanol:acetic acid (95:5) solution was used for 10 min to obtain cell fixation. Preventing non-specific antibody binding using blocking buffer (2% BSA, 5% donkey serum, and 0.1% Triton-X100; Merck Life Sciences Sigma Aldrich, Darmstadt, Germany) was preceded by a double washing with D-PBS (Merck Life Sciences Sigma Aldrich, Darmstadt, Germany). The cells were then incubated overnight at 4 °C with the anti-WWOX/AP-2α/AP-2γ primary antibodies (Thermo Fisher Scientific, Naarden, the Netherlands; catalogue numbers: PA5-29701, MA5-14856 and PA5-17330, respectively) diluted 1:100 in 5% donkey serum. The cells were washed twice with D-PBS and incubated for 1 h with donkey anti-rabbit secondary antibodies (Alexa Fluor 594, catalogue number: A21207; Thermo Fisher Scientific, Naarden, the Netherlands) diluted 1:1000. The ProLong^TM^ Gold antifade reagent containing DAPI (Thermo Fisher Scientific, Naarden, the Netherlands) was used to counterstain the nuclei, and subsequently, the cells were imaged using Eclipse Ci-S (Nikon, Tokyo, Japan) at 40× magnification, 0.75 numerical aperture. The assay was performed in triplicate for each cellular variant.

### 2.5. Assessment of Redox Potential, Proliferation and Apoptosis (Triplex Assay)

The assays investigating differences in cell viability, apoptosis and proliferation between cellular variants were performed on a single plate. The cells were seeded in full medium on a 96-well plate at a concentration of 1.5 × 10^4^ cells per well and incubated for 24 h. The medium was then changed into 100 µL starvation medium containing 10 µL of 5-bromo-2′-deoxyuridine (BrdU) and the cells were subsequently incubated for 24 h. Following this, 10 µL PrestoBlue reagent (Thermo Fisher Scientific, Naarden, the Netherlands) was added to the wells and fluorescence (excitation at 550 nm and emission at 590 nm) were measured at 10 min intervals (control wells: without cells and reagent). Apoptosis and proliferation were detected according to the manufacturer’s protocols using TUNEL assay (DELFIA^®^ DNA Fragmentation Assay; PerkinElmer, Vaughan, ON, Canada) and BrdU incorporation (DELFIA^®^ Cell Proliferation Kit; PerkinElmer, Vaughan, ON, Canada), respectively. The VICTOR X4^TM^ Multilabel Plate Reader (PerkinElmer, Vaughan, ON, Canada) was used to detect fluorescence signals. The assay was performed in triplicate for each cellular variant.

### 2.6. Adhesion Assay

The Corning^®^ BioCoat^TM^ plates (BD Biosciences, San Jose, CA, USA) were used to investigate cell adhesion to collagen I and IV, laminin I or fibronectin with BSA-coated plate as a control. Cellular variants were seeded (1.35 × 10^5^ cells per well) in serum-free media and incubated for 4 h at 37 °C in 5% CO_2_. The cells were then washed three times with 1× PBS (Merck Life Sciences Sigma Aldrich, Darmstadt, Germany) and stained for 10 min using 0.1% crystal violet (Merck Life Sciences Sigma Aldrich, Darmstadt, Germany). Extraction using 10% acetic acid was followed by the absorbance measurement at 560 nm with an Infinite F50 Tecan plate reader (Life Sciences, Mannedorf, Switzerland). The assay was performed in quadruplicate for each cellular variant.

### 2.7. Clonogenic Assay

Cells were seeded (1 × 10^3^/well) onto a 6-well plate in full medium and cultured for 10 days (37 °C, 5% CO_2_) with media exchange every 3 days. All cellular variants were then washed twice with PBS and subjected to fixation with 4% paraformaldehyde (Merck Life Sciences Sigma Aldrich, Darmstadt, Germany) in PBS solution and staining with 0.005% crystal violet (Merck Life Sciences Sigma Aldrich, Darmstadt, Germany) for 15 min at room temperature. The colonies were counted using ImageJ software [25]. The assay was performed in triplicate for each cellular variant.

### 2.8. 3D Culture Growth Assay

Geltrex comprises a basement membrane matrix including laminin, collagen IV, entactin/nidogen and heparin sulfate proteoglycan, which are implicated in the tissue organization. The solidified 2 mm layer of Geltrex matrix (Thermo Fisher Scientific, Naarden, the Netherlands) was added on a 96-well plate prior to seeding cells (1.5 × 10^3^ cells per variant) that were then incubated for 5 days. Subsequently, the cells were observed under the light microscope. This assay was performed in triplicate for each cellular variant.

### 2.9. Invasion Assay

The assessment of invasiveness potential was evaluated using the Corning^®^ BioCoat™ Matrigel^®^ Invasion Chambers with an 8 μm polyester membrane. The full medium was added to the wells and then starvation medium was used to suspend cells which were seeded (2 × 10^5^/well) onto the inner compartment of inserts and incubated (48 h, 37 °C, 5% CO_2_). The cells attached to the membranes outer side were stained with 0.1% crystal violet. Transfer of inserts to 200 μL of extraction solution (10% acetic acid; Merck Life Sciences Sigma Aldrich, Darmstadt, Germany) was followed by incubation on an orbital shaker. The intensity of extracts was measured spectrophotometrically using an Infinite F50 Tecan plate reader (Life Sciences, Mannedorf, Switzerland). The assay was performed in triplicate for each cellular variant.

### 2.10. Gelatin Zymography Assay

Cellular variants were seeded on 6-well plates and cultured in full medium to obtain 80% confluence, after which the medium was exchanged to starving. After 48 h (37 °C, 5% CO_2_), the starvation medium was collected and centrifuged. The Qubit Protein Assay on a Qubit 2.0 Fluorometer (Thermo Fisher Scientific, Naarden, the Netherlands) was used to measure the protein concentration, and subsequently, 3 µg of proteins was used for gelatin-supplemented (2 mg/mL; Merck Life Sciences Sigma Aldrich, Darmstadt, Germany) SDS-PAGE electrophoresis. The gels were washed (2 × 30 min) with 2.5% Triton X-100 (Merck Life Sciences Sigma Aldrich, Darmstadt, Germany), followed by overnight incubation in a developing buffer (0.5 M Tris-HCl, 2 M NaCl, 50 mM CaCl_2_, pH 7.5) at 37 °C. The Coomassie Brilliant Blue R-250 (Merck Life Sciences Sigma Aldrich, Darmstadt, Germany) was used to stain the resulting zymogram and the gel was subsequently washed with destaining solution (methanol:acetic acid:water, 3:1:6). Clear bands over a stained dark blue background appeared at areas of gel where gelatin was enzymatically degraded. The protein ladder was used to confirm the molecular weights of the Matrix Metalloproteinases (MMPs); their activity was estimated using ImageJ software [25]. The assay was performed in triplicate for each cellular variant.

### 2.11. Statistical Analysis

The Shapiro–Wilk test allowed to determine the normality of distribution. An unpaired *t*-test evaluated statistical relevance; results with a *p*-value < 0.05 were considered statistically significant.

### 2.12. RNA Isolation, CAGE Library Preparation, Sequencing, Mapping and Gene Expression

Isolation of RNA was performed according to a protocol provided with Extracol reagent (EURX, Gdansk, Poland) and the quality of total RNA was assessed by Agilent 2100 Bioanalyzer (Agilent Technologies, Santa Clara, CA, USA) to confirm that RNA integrity number (RIN) > 7.0. Synthesis of cDNA from total RNA was accomplished using random primers (CAGE library preparation Kit; K.K. DNAFORM, Kanagawa, Japan). The ribose diols in the 5′ cap of the RNAs were oxidized and biotinylated. Cap-trapping on streptavidin beads allowed selection of RNA/cDNA hybrids. Digestion of RNA by RNaseH preceded the adaptor ligation to both cDNA ends, after which double-stranded cDNA libraries were constructed.

Sequencing of Cap Analysis Gene Expression (CAGE) libraries was performed using single-end reads of 75 nt on a NextSeq 500 instrument (Illumina, San Diego, CA, USA). Quality Check (QC) of fastq files was done using the FastQC analysis tool. Obtained reads/tags were mapped to the human hg38 genome using Burrows–Wheeler Aligner (BWA), version 0.7.17. Unmapped reads were then mapped by Hierarchical Indexing for Spliced Alignment of Transcripts (HISAT2), version 2.0.5.

Tag count data were clustered using the improved Paraclu [26] from RECLU pipeline [27]. Clusters having Counts Per Million (CPM) < 0.1 were rejected. Regions with 90% overlap between replicates were extracted by BEDtools (version 2.12.0). The clusters with Irreproducible Discovery Rate (IDR) ≥ 0.1 and >200 bp were discarded.

### 2.13. Investigation of Differentially Expressed Genes

The edgeR package of the Bioconductor project allowed to find Differentially Expressed Genes (DEGs) with the use of DGEList() constructor and calculating normalization factors to scale the raw library sizes using calcNormFactors() and the normalization method of the Trimmed Mean of M-values (TMM). The most differentially expressed genes were extracted in the form of the table using topTags() ranked by absolute log-Fold-Change (logFC) with *p* < 0.01 adjusted using Benjamini–Hochberg correction method. DEGs were visualized on mean-difference plots using the plotMD() function.

### 2.14. Identification of Target Genes for AP-2 Transcription Factors

List of targets for both AP-2α and AP-2γ were elaborated using databases: Gene Transcription Regulation Database (GTRD, version 19.10 [28]), TRANScription FACtor database (TRANSFAC, version 2019.2) and Transcriptional Regulatory Relationships Unraveled by Sentence-based Text mining (TRRUST, version v2), data status of 8 October 2019. Excluding duplicates, 4810 targets were identified for AP-2α and 5175 for AP-2γ.

### 2.15. Ontological Annotation, Network Visualization and Correlation Analysis

The overrepresentation test from the Protein Analysis Through Evolutionary Relationships (PANTHER) Classification System with Fischer’s exact test was used to annotate the genes to the biological processes that they are implicated in (Annotation Data Set: GO-Slim Biological Process). Available profiles (upregulated or downregulated) of genes were put separately to functional annotation. Visualization of network through Cytoscape software was done using the Search Tool for the Retrieval of Interacting Genes/Proteins (STRING) protein query option, default parameters. Correlation analysis of AP-2 and targets was performed on BLCA tumor dataset from TCGA (data status of 14 February 2021) via the Gene Expression Profiling Interactive Analysis 2 (GEPIA2) database with Spearman’s rank correlation coefficient.

### 2.16. Heatmap Visualization

The freely available shiny web server Heatmapper (http://www.heatmapper.ca/, status of 2 June 2021) was used to generate heatmaps. For CPM, the row scale type was set along with average linkage clustering method and Euclidean distance measurement. Clustering to rows was applied to visualize the dendrogram. For fold-change values, scale type and clustering method was set to “none”.

## 3. Results

### 3.1. Stable Transductions Were Confirmed on the Protein Level

Transductions were evaluated by measuring the relative amounts of WWOX, AP-2α and AP-2γ protein in all variants (Figure 2). The measurements confirmed elevated WWOX expression in “WWOX” (W) variants of both G3 (HT-1376) and G4 (CAL-29) bladder cancer cell lines (mean expression level 7.109 ± 1.049 and 5.325 ± 0.491, respectively) compared to “contr” (K) variants of both cell lines (mean expression level 1.776 ± 0.259, *p* = 0.0010 and 1.109 ± 0.079, *p* < 0.0001, respectively), thus confirming the first stable transduction. Subsequently, both K/A vs. K/K and W/A vs. W/K comparisons revealed successful second transduction regarding AP-2α. For HT-1376, mean expression level 0.454 ± 0.010 for K/A vs. mean expression level 0.036 ± 0.006 for K/K (*p* = 0.0004) and mean expression level 0.115 ± 0.004 for W/A vs. mean expression level 0.043 ± 0.014 for W/K (*p* = 0.0211). Considering CAL-29, mean expression level 0.854 ± 0.003 for K/A vs. mean expression level 0.027 ± 0.005 for K/K (*p* < 0.0001) and mean expression level 0.831 ± 0.039 for W/A vs. mean expression level 0.039 ± 0.008 for W/K (*p* = 0.0013). The second transduction in regard to AP-2γ was confirmed analogously. For HT-1376, expression of AP-2γ in the K/C variant (mean expression level 4.476 ± 0.814) was greater than in K/K (mean expression level 0.065 ± 0.006, *p* = 0.0166) and W/C (mean expression level 1.468 ± 0.045) was greater than W/K (mean expression level 0.076 ± 0.001, *p* = 0.0005). The same was confirmed in the case of CAL-29, mean expression level 1.065 ± 0.024 for K/C vs. mean expression level 0.034 ± 0.002 for K/K (*p* = 0.0003) and mean expression level 1.424 ± 0.137 for W/C vs. mean expression level 0.031 ± 0.001 for W/K (*p* = 0.0048).

### 3.2. WWOX, AP-2α and AP-2γ Were Diversely Localized Depending on the Cellular Variant

Immunofluorescence shows that in variants demonstrating overexpression of AP-2α or AP-2γ versus their control, AP-2α is located solely in the nucleus and WWOX mainly in the cytoplasm; however, AP-2γ might be localized in both the nucleus and cytoplasm. It was more emphasized in the CAL-29 cell line, but a subtle manifestation was also present in HT-1376 (Figure 3A, variant K/C anti-AP-2γ, upper left corner). In the W/A and W/C variants of both cell lines, WWOX was again captured primarily in the cytoplasm while AP-2 factors were scattered in the cell nucleus and cytoplasm. Notably, apart from the similarities in WWOX, which are consistent with the above observations, in W/K, the AP-2α appeared to be localized in the cytoplasm while AP-2γ remained in the nucleus. All described data are visualized in Figure 3.

### 3.3. WWOX and AP-2α Functionality Coincided Together But Contradicted with AP-2γ

Programmed cell death, proliferation and mitochondrial redox potential were investigated together using Triplex assay. Apoptosis (Figure 4A,B) was increased in K/A (1.25-fold, *p* = 0.0351) of HT-1376 cells, the same in W/K of HT-1376 (1.58-fold, *p* = 0.0005) and CAL-29 (1.94-fold, *p* = 0.0081). In contrast, K/C decreased apoptosis (1.25-fold, *p* = 0.0212) as well as W/C (1.51-fold, *p* < 0.0001) in HT-1376 cells. Likewise, W/A was anti-apoptotic (1.38-fold, *p* = 0.0429) in CAL-29.

In terms of proliferative potential (Figure 4C,D), the only comparable variant between cell lines was K/A, which decreased proliferation in both cases (1.24-fold, *p* = 0.0085 for HT-1376 and 1.36-fold, *p* = 0.0040 for CAL-29). Based on the HT-1376 cells, the proliferation was increased in K/C (1.15-fold, *p* = 0.0188), decreased in W/K (1.72-fold, *p* < 0.0001), decreased in W/A (1.69-fold, *p* = 0.0002) and increased in W/C (1.24-fold, *p* = 0.0147).

Mitochondrial redox potential indicates cell viability; this assay was analyzed as described previously [8], i.e., the variant was considered meaningful if at least three measurement points met statistical significance. Thus, HT-1376 cell viability decreased after WWOX overexpression (mean 1.12-fold) but increased after AP-2γ overexpression (mean 1.16-fold). In both K/C and W/C of CAL-29 cells, the viability was increased by 1.56-fold and 1.29-fold on average. The remaining variants in G4 cells were K/A and W/A, both decreased redox potential (mean 1.43-fold for K/A and mean 1.57-fold for W/A). Mitochondrial redox potential of Triplex assay is visualized in Figure 5.

### 3.4. Individual Overexpression of AP-2γ Increased the Adhesion to Each Analyzed ECM Protein in Both G3 and G4 BLCA

The statistically significant comparisons varied depending on the cell line (Figure 6). Adhesion to all Extracellular Matrix (ECM) proteins in W/A or W/C variants of HT-1376 was similar to their control variant i.e., W/K. The other comparisons revealed that overexpression of AP-2α (K/A) decreased adhesion, while AP-2γ (K/C) increased it to all investigated ECM proteins: collagen I (1.35-fold, *p* < 0.0001 for K/A and 1.49-fold, *p* < 0.0001 for K/C), collagen IV (1.41-fold, *p* < 0.0001 for K/A and 1.56-fold, *p* < 0.0001 for K/C), laminin I (1.27-fold, *p* = 0.0005 for K/A and 1.82-fold, *p* < 0.0001 for K/C) and fibronectin (1.64-fold, *p* < 0.0001 for K/A and 1.54, *p* < 0.0001 for K/C). WWOX overexpression (W/K) decreased adhesion to both collagens (1.12-fold, *p* = 0.0122 for collagen I and 1.29-fold, *p* < 0.0001 for collagen IV) and laminin I (1.15-fold, *p* = 0.0058) but not to fibronectin.

Similar relationships were observed in HT-1376 as in CAL-29, although with additional remarks. Briefly, the same conclusion can be drawn from K/A and K/C; however, AP-2α overexpression did not significantly decrease adhesion to laminin. Thus, AP-2γ overexpression (K/C) was the only variant which demonstrated comparable activity for all ECM proteins in both cell lines; in CAL-29 it potentiated adhesion to collagen I by 1.34-fold (*p* < 0.0001), collagen IV by 1.36-fold (*p* < 0.0001), laminin I by 1.57-fold (*p* = 0.0008) and fibronectin by 1.31-fold (*p* = 0.0116) compared to K/K. The K/A variant to both collagens (1.37-fold, *p* < 0.0001 for collagen I and 1.26-fold, *p* < 0.0001 for collagen IV) and fibronectin (1.62-fold, *p* = 0.0003), while W/K decreased adherence only to collagen I by 1.11-fold (*p* = 0.0226), both compared to K/K. The results for CAL-29 indicated that W/A variant decreased the adhesion to all investigated ECM proteins, while W/C increased it (collagen I: W/A 1.56-fold and *p* < 0.0001, W/C 1.26-fold and *p* < 0.0001; collagen IV: W/A 1.31-fold and *p* < 0.0001, W/C 1.53-fold and *p* < 0.0001; laminin I: W/A 1.17-fold and *p* = 0.0045, W/C 1.62-fold and *p* < 0.0001; fibronectin: W/A 1.42-fold and *p* < 0.0001, W/C 1.65-fold and *p* < 0.0001).

### 3.5. Self-Renewal Capacities and Dimensional Growth Varied Depending on the WWOX, AP-2α and AP-2γ Level

Both clonogenicity and 3D culture growth revealed prominent differences between variants. The AP-2α overexpression decreased the ability of a single cell to grow into colony for HT-1376 by 1.35-fold, *p* = 0.0199 (Figure 7A) and CAL-29 by 3.42-fold, *p* = 0.0004 (Figure 7B). The changes were also noticeable in Geltrex basement membrane extract (Figure 7C), where high AP-2α decreased growth; this effect was more visible in the quantity of colonies for HT-1376 but in sphere size for CAL-29. Similar to AP-2α, WWOX overexpression decreased colony formation for both cell lines (1.77-fold, *p* = 0.0010 and 2.65-fold, *p* = 0.0004 for both HT-1376 and CAL-29, respectively), with fewer total (HT-1376) or smaller (CAL-29) colonies growing in 3D culture. In contrast, more colonies were observed when AP-2γ was overexpressed (1.31-fold, *p* = 0.0373 and 1.80-fold, *p* = 0.0044 for both HT-1376 and CAL-29, respectively), with a higher level of dimensional growth in CAL-29 (more colonies, larger spheres) than HT-1376 (slightly more colonies, no change in size of the cells’ clusters). Overexpression of both WWOX and AP-2α (W/A) resulted in no noticeable differences in spatial growth of either HT-1376 or CAL-29 cells. However, it demonstrated lower clonogenicity than W/K (2.09-fold, *p* = 0.0001 and 1.85-fold, *p* = 0.0011 for both HT-1376 and CAL-29, respectively), suggesting possible synergistic effect between WWOX and AP-2α. Finally, W/C increased colony formation (1.41-fold, *p* = 0.0479) and spatial growth of HT-1376 (more colonies, single ones forming larger spheres), while the effect is limited only to larger spheres formation in CAL-29 cells.

### 3.6. All Cellular Variants with AP-2γ Overexpression Increased Invasiveness Potential

The results of gelatin zymography assay in HT-1376 were statistically significant for all possible comparisons regarding the MMP-9 activity; no MMP-2 activity was observed. This shows that the K/A variant inversely regulated MMP-9 (1.64-fold decrease, *p* = 0.0022), as opposed K/C (Figure 8A), which increased the metalloproteinase activity (4.42-fold, *p* = 0.0003) similarly to W/K (5.16-fold, *p* = 0.0005). Furthermore, MMP-9 activity in W/A and W/C corresponded to K/A and K/C; it was decreased in the first one (1.66-fold, *p* = 0.0100) but increased in the second (1.70-fold, *p* = 0.0014). The results for the CAL-29 cell line (Figure 8B) were coherent with HT-1376 in the K/C and W/C variants that increased MMP-9 activity compared to controls (13.09-fold, *p* < 0.0001 or 2.99-fold, *p* = 0.0023, respectively). The opposite was observed for MMP-2 activity (Figure 8C) but it was more comparable with HT-1376 due to the large number of significant comparisons (for CAL-29, MMP-9 changes were significant only for K/C and W/C, Figure 8B). For both K/A and K/C, the trend in MMP-2 regulation (Figure 8C) was contrary to MMP-9 (Figure 8A) i.e., K/A increased while K/C decreased MMP-2 activity (2.67-fold, *p* = 0.0015 or 2.29-fold, *p* = 0.0291, respectively). However, W/K maintained its capabilities to increase MMP-2 by 1.69-fold (*p* = 0.0159, Figure 8C) similarly to MMP-9 (Figure 8A).

The invasiveness assay was then compared with the results from zymogram. Consistently, all variants with AP-2γ overexpression for both cell lines (Figure 8D,E) increased the motility potential (1.83-fold, *p* < 0.0001 for K/C HT-1376; 1.72-fold, *p* = 0.0004 for W/C HT-1376; 2.45-fold, *p* < 0.0001 for K/C CAL-29; 1.75-fold, *p* = 0.0004 for W/C CAL-29). Overexpressed AP-2α appeared to increase the invasion of CAL-29 cells (Figure 8D) in both K/A (1.32-fold, *p* = 0.0144) and W/A (1.30-fold, *p* = 0.0130) variants. WWOX overexpression did not affect the migratory potential in any cell line (Figure 8D,E).

### 3.7. CAGE Passed Quality Check

In vitro assays were followed by high-throughput sequencing, which was used to determine DEGs between variants. Generated per-base sequence quality plots showed the average Phred scores according to the sequence reads’ length (Appendix A); these were of good quality, as no raw reads exceeded the green background of the plots and the mean scores were >28. The remaining background colors, i.e., orange or red, represented a base calling of reasonable or poor quality, while a blue line indicated the mean score. Processed CAGE data as CPM of genes are merged in Appendix A.

### 3.8. Many Differentially Expressed Genes Were Found among Cellular Variants

Further investigation indicated how many DEGs are identified between two variants i.e., K/A vs. K/K, K/C vs. K/K, W/K vs. K/K, W/A vs. W/K or W/C vs. W/K. DEGs were considered significant when at least logFC > 1.5 and *p* < 0.01. The genes were first filtered by an adjusted *p*-value using the topTags() function and visualized through Mean-Difference (MD) plots (Figure 9). Following this, logFC was applied to screen genes from topTags() to select DEGs. In some comparisons, all genes already had logFC > 1.5, yet in others, it yielded a limited set of genes. Altogether, about 2500 DEGs (excluding duplicates: more than 1500) were found for both cell lines and from all comparisons; these are collected in Appendix A.

### 3.9. Gene Ontology Revealed Processess That Underlie Several Cancer Hallmarks

Subsequently, DEGs were implemented into PANTHER for functional annotation. Differences between K/A and K/K (1) for CAL-29 were related to biological processes e.g., apoptosis, adhesion, protein degradation, migration and cell cycle, but also signaling pathways mediated by ERBB4, NOTCH4, RAS or WNT; (2) for HT-1376 there were many repetitions; however, ECM organization, cell proliferation, angiogenesis or NFκB signaling, among others, did not appear in CAL-29. Henceforth, “(1)” will be used to present descriptors of CAL-29 variants, while “(2)” will precede subsidiary data from HT-1376 comparisons. Regarding K/C vs. K/K, there were changes in (1) cell death, migration, growth, blood vessels morphogenesis, integrin interactions, collagen degradation, ECM organization, G_2_/M cell cycle checkpoint, oncogene-induced senescence, ERBB and TGFβ pathways; (2) cell proliferation, adhesion, wound healing, angiogenesis, Epithelial-to-Mesenchymal Transition (EMT), anoikis, SMAD and MAPK signaling. For W/K vs. K/K, there were alterations in: (1) programmed cell death, angiogenesis, mitotic cell cycle, adhesion, migration, proliferation, NOTCH and RAF/MAPK signaling; (2) EMT, cell growth and size, ECM organization, TGFβ and WNT pathways. Furthermore, W/A and W/K differed in terms of: (1) cell cycle, apoptosis, proliferation, wound healing, ECM organization, EGFR and NOTCH pathways; (2) angiogenesis and signaling of SMAD, TGFβ, MAPK or WNT. Lastly, W/C and W/K varied in: (1) proliferation, cell cycle, ECM organization and ERBB, TGFβ and WNT pathways; (2) inflammatory response, apoptosis, adhesion and NFκB signaling. The details of the ontological analysis (along with candidate DEGs and statistical significance) are collected in Appendix A.

Overall, the processes were in line with cancer hallmarks, i.e., sustaining proliferative signaling (cell cycle, proliferation), resisting cell death (apoptosis) or deregulating angiogenesis. Moreover, processes related to activating invasion and metastasis (migration, EMT, ECM organization) were also very common. We believe these were, at least to a certain extent, regulated by the range of signaling pathways mentioned above.

### 3.10. The Most Frequent DEGs Included BLCA-Related Genes and AP-2 Targets

The most frequent genes were included in the next step if they have been annotated in at least half of all possible comparisons. DEGs were also screened to indicate if they are targets of AP-2 transcription factors according to the GTRD, TRANSFAC and TRRUST databases.

In total, 21 genes met the requirement of frequency: *ADIRF*, *AREG*, *ATOH8*, *CCL2*, *CD74*, *CFB*, *FOSL1*, *GM2A*, *H3C3*, *HPGD*, *KLK7*, *KLK8*, *MYCL*, *PSCA*, *S100A2*, *S100A7*, *S100A8*, *S100A9*, *TGM2*, *TNC* and *UBD* (Appendix A). Among them, *ATOH8*, *GM2A*, *PSCA*, *TGM2* and *TNC* were found as targets of both AP-2α and AP-2γ, while *FOSL1* and *S100A2* were exclusively AP-2α targets and *MYCL* was a target of AP-2γ. The S100 family genes were problematic, as *S100A7/A8/A9* genes were not found as targets; however, both AP-2 factors regulate *S100A5/A11/A13/P* genes and a few others separately, which might suggest that a broader view is needed to understand the regulation of S100 genes by AP-2. The remaining genes were not found on lists acquired from GTRD/TRANSFAC/TRRUST. 

Subsequently, all genes were implemented to Cytoscape to find whether they might present a network of protein–protein interactions. This is presented in Figure 10, together with a few acceptable correlations of AP-2 and its target in BLCA. As more than half of the genes interacted, the analysis was focused on examining their potential role in bladder cancer. The summary in Table 1 indicates that a majority of them had BLCA-related functions or served as potential biomarkers; their expression in cellular variants is presented in Figure 11.

## 4. Discussion

*WWOX* is a haploinsufficient gene whose downregulation has been associated with a higher BLCA grade [22,42]. The loss of its expression in bladder cancer can be induced by hypermethylation of the promoter or first exon, which is caused by tobacco smoking [5], a recognizable risk factor for BLCA patients [6]. AP-2α and AP-2γ have been already linked to carcinogenesis (including BLCA [15,17]) but demonstrate opposite effects, e.g., AP-2α is downregulated during tumor genesis while AP-2γ potentiates cancer progression [8]. Our previous study paved the way for understanding the WWOX–AP-2 axis in BLCA and indicated how it affects biological processes. However, the spectrum of such findings was restricted to G2 bladder cancer; hence, the current study aimed to investigate this issue in more aggressive cancers with higher grades.

Immunocytochemistry revealed that overexpression of a particular gene resulted in their accumulation in the most characteristic cellular compartment: the cytoplasm for WWOX and the nucleus for both AP-2 TFs. This was consistent in the case of WWOX (W/K vs. K/K) and AP-2α (K/A vs. K/K), but not for AP-2γ (K/C vs. K/K), which was located in both the nucleus and cytoplasm. Such observations are different from previous research, where AP-2γ was localized in the nucleus regardless of the expression status and was only altered due to sequestration by WWOX [15]. Although no literature data concern AP-2γ, Li et al. found oncogenic AP-2β to be synchronously localized in both nucleus and cytoplasm in breast cancer patients with high AP-2β expression; no cytoplasmic distribution was observed in patients with low AP-2β [43]. Considering the similar nature of AP-2γ, this relocation might be attributed to overexpression, serving as a purposeful mechanism of tumor development.

Overexpression of WWOX together with AP-2α (W/A) or AP-2γ (W/C) indicated that all these molecules might be localized in the nucleus and cytoplasm, which corresponds to our previous research. However, in both cell lines, the WWOX overexpression, either with (W/A) or without AP-2α overexpression (W/K), showed localization of AP-2α in cytoplasm. This confirms the ability of WWOX to sequestrate AP-2α out of the nucleus, since in K/K or K/A, only nuclear signals were found in the case of anti-AP-2α. Contrastingly, WWOX overexpression alone (W/K) did not translocate AP-2γ; this suggests a WWOX-independent location of AP-2γ in both the nucleus and cytoplasm, which is also visible in K/C. The changing compartment was not surprising, as WWOX is known to sequestrate other proteins [44], yet the occurrence of such translocation independent on WWOX (particularly in K/C) was intriguing.

The results of WWOX–AP-2γ collaboration differ from those observed previously in G2. Overexpression of AP-2γ (in both K/C and W/C) increased tumor cell viability and proliferation, but decreased programmed cell death. Such AP-2γ nature might be related to c-MYC inhibition (to suppress apoptosis) or an increase of EGFR (to intensify proliferation) [45,46]. However, the impact on the mitochondrial redox potential needs reference to AP-2β, whose knockdown suppressed the cell viability in breast cancer [43]. Interestingly, WWOX overexpression in W/C did not alter AP-2γ function, but its separate overexpression (W/K) shows an anti-cancer effect via the induction of apoptosis or inhibition of cell viability and proliferation. Similar results have been observed in another BLCA study [47], but also in colon, prostate, pancreatic and lung cancers [48,49], and might be related to enhanced TNFα, which induces cell death and decreases viability [50,51]. Proliferation was found to be increased when *WWOX* gene was methylated in osteosarcoma [52], while apoptosis was potentiated in ovarian carcinoma through caspase-3 and PARP when WWOX was functional [53]. Moreover, the impact of WWOX on proliferation has also been confirmed in bladder cancer [47]. The anti-proliferative ability of WWOX in BLCA can be explained using ERBB4, a receptor that plays a crucial role in bladder carcinogenesis [54]. Briefly, WWOX impairs proliferation by preventing the translocation of ERBB4′s intracellular C-terminal domain into the nucleus [55]. Separate overexpression of AP-2α (K/A) or with WWOX (W/A) mainly exhibited anti-cancer properties, i.e., increased apoptosis or decrease of both cell viability and proliferation; however, there was one disparity in the W/A of the CAL-29 cells. Namely, it appeared to decrease apoptosis but it does not seem to be WWOX-dependent, as WWOX increased programmed cell death in the same cells. This may be due to AP-2α; however, such an observation is ambiguous, since its separate overexpression in the same cells decreased the proliferative potential, suggesting that it can hinder cancer progression. This is in line with a study by Zeng et al., who indicated that AP-2α can inhibit proliferation and activate apoptosis of gastric cancer cells through the regulation of ERBB2, ERα, p21 and caspase-3, -8 and -9 [56]. Moreover, Huang et al. reported that AP-2α suppressed hepatocellular carcinoma survival via the downregulation of, e.g., ERK, CCND1 or CD133 [57], the latter having a significant influence on BLCA viability [58]. In brief, the results of the Triplex assay indicate that WWOX and AP-2α are likely to synergize, but they act inversely to AP-2γ.

Investigation of adhesive properties emphasized similar observations. In bladder cancer, reduction of cellular adhesion was followed by decreased aggressiveness [59]; in other tumors, the attachment of cells to the ECM promotes proliferation, chemoresistance or survival [60,61]. Thus, decreased adhesion may prevent cancer development; in the present study, only variants with AP-2γ (K/C, W/C) increased adhesion to ECM proteins. Such an effect might depend on AP-2γ, as both WWOX and AP-2α decreased adhesion to a specific ECM protein, if statistically significant. In few cases, a synergistic effect was required between WWOX and AP-2α to demonstrate the adhesiveness reduction. The example of endometrial carcinoma revealed that integrin β4 and α2 are upregulated after WWOX silencing [62], suggesting that WWOX may regulate adhesion to laminin and collagens, respectively [63]. This is in line with another study on BLCA, where WWOX silencing increased adhesion to collagen IV [47]. On the other hand, AP-2γ was found to remodel the extracellular matrix during carcinogenesis via RND3 in a process known as ECM anisotropy, which involves fibronectin [64]. Lastly, our previous study on G2 BLCA proposed an α2 integrin-encoding gene as the AP-2α target [8].

Proceeding with in vitro experiments, we noticed that both WWOX and AP-2α frequently synergized and yielded different results than AP-2γ. Clonogenicity assay presented that either separate WWOX (W/K) or AP-2α (K/A) overexpression reduced colony formation; their collaboration (W/A) decreased it even further. This fits previous findings that WWOX inhibits the clonogenicity of bladder cancer [47] but also pancreatic carcinoma via triggering apoptosis in a p53-dependent mechanism [65] or leukemia by the release of cytochrome c [66]. Likewise, AP-2α was found to activate p21, which inhibited DNA synthesis and the cell cycle, decreasing cell growth [67]. Regarding AP-2γ, its individual overexpression (K/C) increased the number of colonies of both cell lines, but when combined with WWOX (W/C), it only appeared to do so in G3 BLCA. This oncogene can increase colony formation due to enhancement of RET signaling via interaction with PELP1 [68].

The 3D culture assay supported the clonogenicity findings. WWOX or AP-2α overexpression decreased the quantity of cellular clusters or their size, while their combined variant (W/A) maintained the observed tendency. In contrast, high AP-2γ expression (K/C) increased both the size and the number of spheres; the net effect of the W/C variant was similar. In our previous paper, we comprehensively discussed in what way WWOX and two AP-2 members differ and how this led to various phenotypes in spatial growth. There appeared to be a dissimilarity in the regulation of Mesenchymal-to-Epithelial Transition (MET)-EMT,a which is implicated in cancer stemness; WWOX is thought to be an EMT inhibitor, AP-2γ an EMT enhancer, while AP-2α is thought to be an MET inducer. The molecules involved in this complex network were collectively TGFBR1, PAK1, MAPK, IGF1R, TNPO1, FASN, TGFB1, c-MYC, ELF5 and SNAI1 [8].

Moving to the last two biological assays, AP-2α overexpression decreased the MMP-9 activity in G3, which is in line with colon cancer findings [69], and increased MMP-2 in G4, as observed by Kuckenberg et al. [70]. The decrease in MMP-9 was also maintained in the W/A variant, but the final effect could be due to AP-2α, which might override the WWOX function in the context of MMP regulation [8]. In contrast, overexpression of AP-2γ increased MMP-9 but decreased this can be referred to study on oncogenic AP-2β whose elevated expression led to MMP-9 upregulation and greater invasiveness [43]. Regarding MMP-2, a similar effect was found in a previous study on G2 BLCA, but only in K/C, as the phenotype of W/C was then dependent on WWOX rather than AP-2γ [8]. The above observations suggest that AP-2α and AP-2γ have opposite effects on MMP-2 and MMP-9 activity. Regarding WWOX, its overexpression upregulated both MMPs, but did not affect BLCA invasiveness; the literature data are disputable, since, depending on the source, it appears to either reduce cellular invasion [71] or not affect it [47,72]. For two AP-2 members, the increased invasiveness observed during AP-2γ overexpression might be due to increased MMP-9 activity, while increased invasion in AP-2α variants (only in G4) may result from MMP-2. Hence, at the later phases of cancer development, AP-2α does not entirely resemble a tumor suppressor; some studies have already questioned its anti-cancer role in pancreatic, nasopharyngeal, squamous cell carcinomas or neuroblastoma and leukemia [9]. Likewise, its impact on invasion is ambiguous [73,74], while AP-2γ apparently acts as an invasiveness enhancer [75,76].

The usage of sequencing data revealed that more than 1500 DEGs distinguished all variants, from about ~180 to up to ~700 genes in specific comparison. The in vitro experiments such as zymography or adhesion via changes in MMPs or integrins (ITGs), were certified in CAGE. For CAL-29 comparisons, MMP-9 increased 9-log-fold in W/C compared to W/K, which is consistent with the zymography assay. In terms of HT-1376, the same metalloproteinase was found to be 6.7-log-fold higher in W/K compared to K/K, which confirms the zymogram findings. Integrins were assessed on the basis of their diversity and ECM-related interactions [63]. Genes encoding integrin β6 or α2 (*ITGB6* or *ITGA2*, respectively) were found to be decreased 2- or 1.8-log-fold in K/A vs. K/K of CAL-29 cells, which could explain the lower adhesive properties for fibronectin or collagens. Similarly, *ITGA2* expression was 2-log-fold reduced in W/K compared to K/K, explaining the reduced adhesion to collagen I. Lastly, a significant increase in the integrin α5-encoding gene (*ITGA5*) was found in W/C with a 1.44-log-fold increase, suggesting greater adhesiveness to fibronectin. The same gene was 1.53-log-fold higher in K/C of HT-1376 cells. ITGs that were repetitive across comparisons and might represent a common thread of regulation consisted of *ITGA2* for WWOX (W/K) and AP-2α (K/A) overexpression and *ITGA5* for variants with AP-2γ overexpression (K/C and W/C). According to GTRD/TRANSFAC/TRRUST databases, *ITGA2* and *ITGA5* are targets of both AP-2 factors. In addition, WWOX has been found to reduce integrin α2 expression [77]. 

Further gene ontology on DEGs revealed that processes related to cancer hallmarks were frequently annotated; namely, cell cycle, proliferation, apoptosis, ECM organization, angiogenesis, EMT or migration. Many pathways appeared in at least three comparisons; MAPK, TGFβ, WNT, NOTCH or ERBB signaling were found to be dependent on WWOX, AP-2α and AP-2γ, i.e., MAPK [78,79,80], TGFβ [80,81,82], WNT [83,84], NOTCH [85,86,87] and ERBB [88,89]. Since WWOX is considered a global gene expression modulator [62] and both AP-2 transcription factors regulate thousands of genes and biological processes via specific pathways [79], it was foreseeable that changes in their expression affect cellular events at the broad scale. CAGE data were filtered to indicate the most frequent DEGs; these genes were also screened as putative AP-2 targets. Several genes from the S100 family were included in the list, suggesting that AP-2 might be involved in S100 proteins regulation; this has been noted in other studies that have focused on specific members [90,91]. Out of 21 genes visualized as network, *S100A2*, *GM2A* and *PSCA* were fairly correlated with AP-2 factors; the latter was not included in Table 1 due to a lack of interaction with other DEGs. Nevertheless, *PSCA* expression was found to be 5.7-fold higher in BLCA when compared with normal bladder tissue [92] and was correlated with increasing tumor grade, which may be of use in BLCA therapy and diagnosis [93]. The negative correlation of *PSCA* and AP-2α suggests that AP-2α may hinder bladder cancer development, since *PSCA* is involved in metastasis and tumor growth, as exemplified through prostate cancer [93]. Collectively, *AREG*, *CCL2*, *CD74*, *CFB*, *FOSL1*, *GM2A*, *KLK7*, *KLK8*, *PSCA*, *S100A2*, *S100A7*, *S100A8* and *S100A9* may be DEGs worth exploring; some of them were not investigated as BLCA biomarkers and are considered novel.

As aforementioned, the concerns about AP-2α function were observed only in G4 regarding apoptosis and invasion. However, dedicated research is needed to elaborate the specific cause of such phenomena since AP-2γ/AP-2α heterodimers [94] were found to modulate binding to target promoters differently than their homodimers [70]. Moreover, increased invasion and decreased apoptosis might not be sufficient to counteract decreased proliferation, viability, adhesion, MMP-9 activity, clonogenicity and spatial growth, which are also observed during AP-2α overexpression.

In terms of AP-2γ, its oncogenic nature is now evident even when accompanied by WWOX (W/C), as assessed by all in vitro assays; previously, such a condition was present in only one assay on G2 BLCA [8]. This could suggest a grade-dependent shift in the superiority of function (AP-2γ over WWOX) or partnership switching. For instance, the best-known competitor of WWOX is YAP, an oncogene which regulates genes similar to AP-2γ [95], suggesting that these two may have more in common. YAP plays a significant role in BLCA, i.e., induces tumor growth and invasion [96], regulates BLCA stemness [97], protects from cancer therapy [98] and correlates with histologic grade [99]. It is also activated through cigarette smoking in head-and-neck, esophageal and cervical carcinomas [100,101], but smoking itself is also a crucial habit among BLCA patients. The fact that WWOX and YAP compete for ERBB4 [88] suggests that in higher BLCA grades, such competition might occur between WWOX and YAP, but in terms of AP-2γ. Considering the oncogenic nature of AP-2γ, the loss of WWOX and overexpression of YAP in high BLCA grade, this might represent a partnership switch from WWOX–AP-2γ to YAP–AP-2γ. Theoretically, AP-2γ could be a potential YAP target, since its Proline-rich (PPxY) motif is recognized by the tryptophan (W) domain of YAP [102], the same as in the WWOX–AP-2γ interaction [103]. Certainly, this might be one of the possibilities and it should be confirmed in the future.

## 5. Conclusions

To conclude, this research confirmed previous data regarding the tumor suppressor nature of WWOX and AP-2α, although the latter appeared to guide singular processes of G4 BLCA in an oncogene-resembling manner. Such properties of AP-2α have been suggested in the literature, and this may be due to the influence of AP-2γ; further examination of the proportions of AP-2 homodimers and heterodimers is advisable. Moreover, AP-2γ appears to be oncogenic regulator of BLCA, which is now more evident through the progressive deregulation of the WWOX–AP-2γ interaction, affecting a multitude of biological processes. In summary, the tumor suppressor synergism between WWOX and AP-2α is unimpaired in high-grade BLCA compared to G2, yet the ability of WWOX to guide oncogenic AP-2γ is almost utterly lost. These alterations may be due to a change in the molecule superiority or partnership switch; therefore, subsequent investigation is needed. In addition, transcriptome profiling of cellular variants by CAGE-seq revealed a promising set of genes (including AP-2 targets) that may be worth exploring in the context of novel BLCA biomarkers.

## Figures and Tables

**Figure 1 cancers-13-02957-f001:**
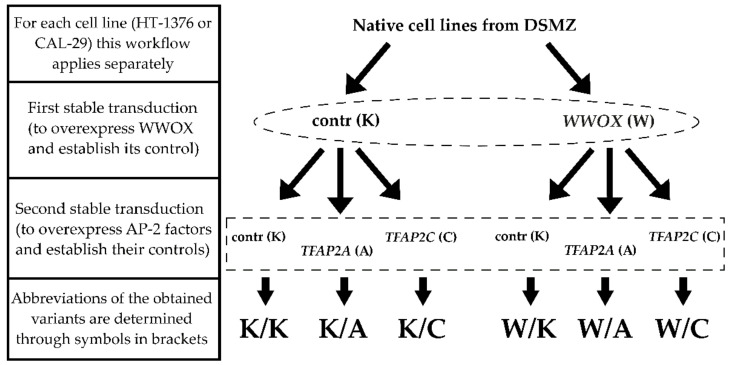
Stable transductions scheme for both HT-1376 and CAL-29 cell lines.

**Figure 2 cancers-13-02957-f002:**
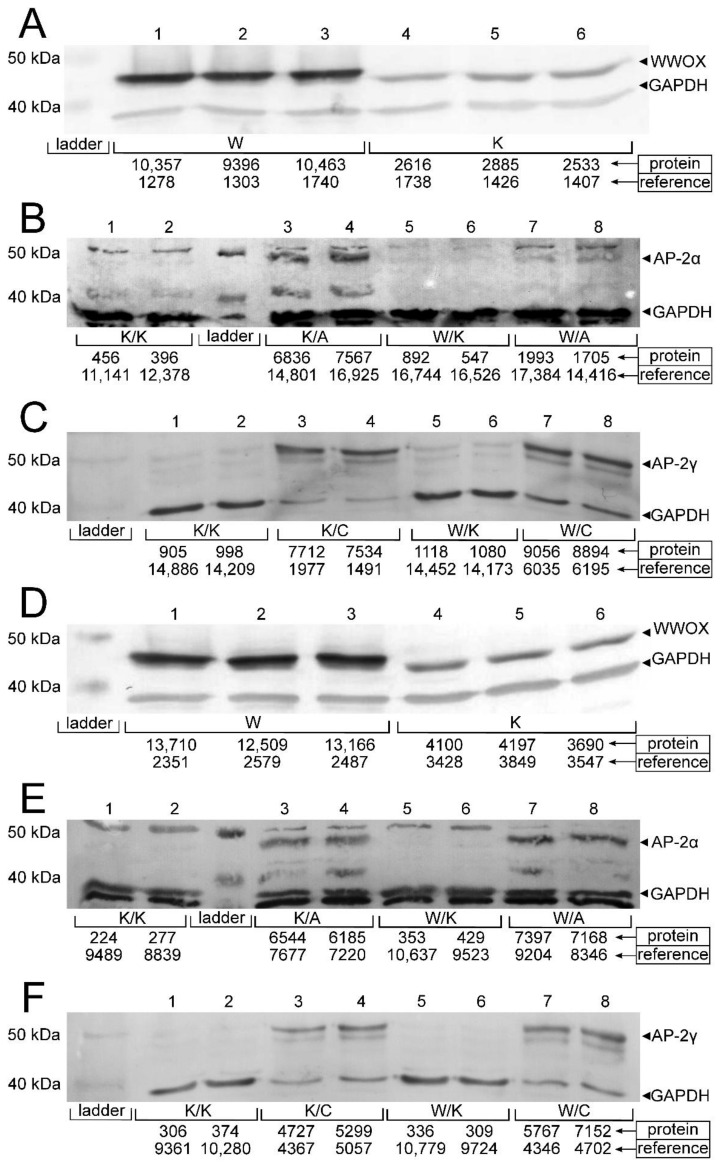
Verification of gene overexpression on the protein level. (**A**) WWOX overexpression in HT-1376. (**B**) AP-2α overexpression in HT-1376. (**C**) AP-2γ overexpression in HT-1376. (**D**) WWOX overexpression in CAL-29. (**E**) AP-2α overexpression in CAL-29. (**F**) AP-2γ overexpression in CAL-29.

**Figure 3 cancers-13-02957-f003:**
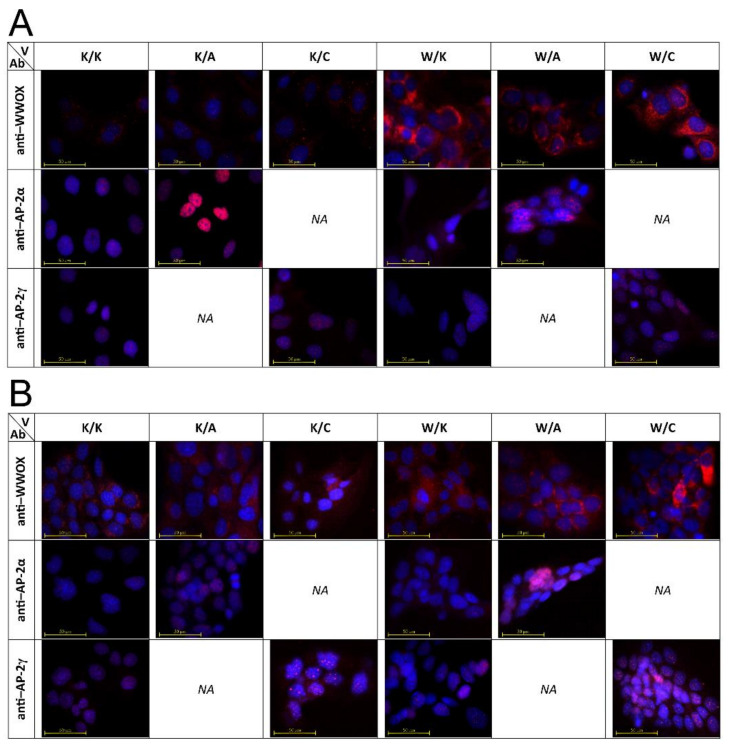
Cellular location of WWOX, AP-2α and AP-2γ within in vitro variants. (**A**) HT-1376 cells. (**B**) CAL-29 cells. V—Variant. Ab—antibody.

**Figure 4 cancers-13-02957-f004:**
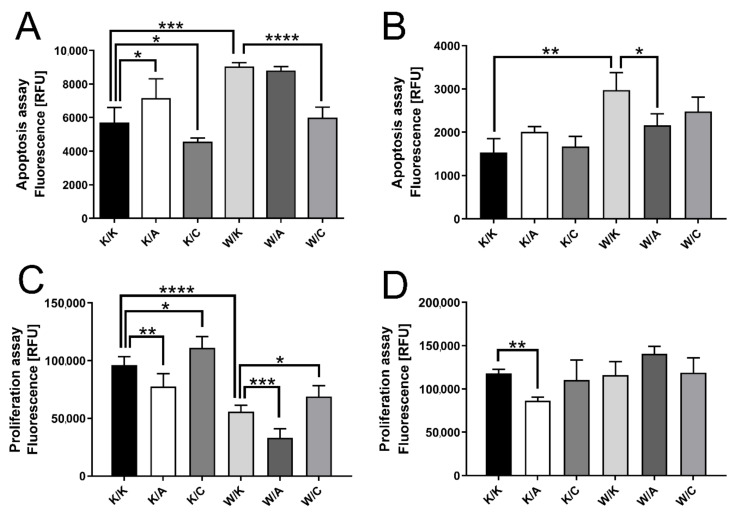
Differences between apoptotic and proliferative potential. (**A**) Apoptosis in HT-1376. (**B**) Apoptosis in CAL-29. (**C**) Proliferation in HT-1376. (**D**) Proliferation in CAL-29. *p* < 0.05 (*), *p* < 0.01 (**), *p* < 0.001 (***), *p* < 0.0001 (****).

**Figure 5 cancers-13-02957-f005:**
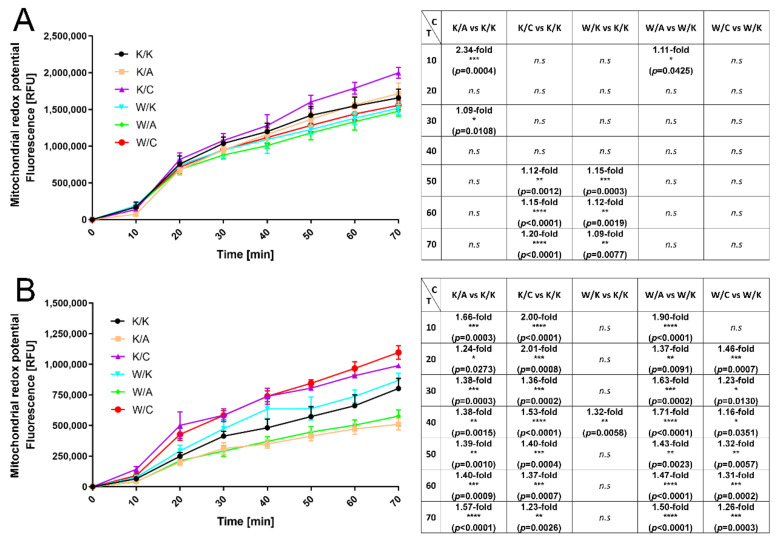
Evaluation of changes in mitochondrial redox potential (graphical and tabular). (**A**) HT-1376 variants. (**B**) CAL-29 variants. C—Comparison. T—Time [min]. *p* < 0.05 (*), *p* < 0.01 (**), *p* < 0.001 (***), *p* < 0.0001 (****).

**Figure 6 cancers-13-02957-f006:**
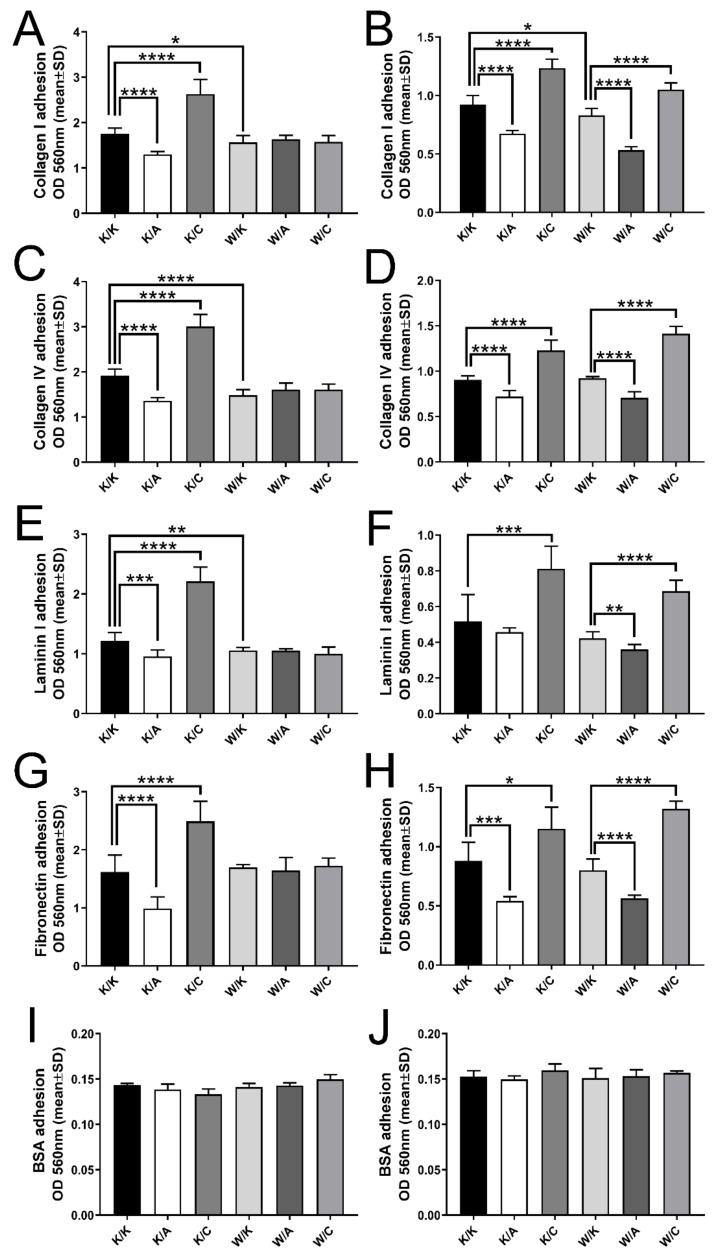
Variability of adhesiveness to ECM proteins. (**A**,**B**) Collagen I. (**C**,**D**) Collagen IV. (**E**,**F**) Laminin I. (**G**,**H**) Fibronectin. (**I**,**J**) BSA (negative control). The results are shown in pairs, i.e., A–B, C–D, E–F, etc.; the paired graphs present data of HT-1376 and CAL-29 cells, respectively. *p* < 0.05 (*), *p* < 0.01 (**), *p* < 0.001 (***), *p* < 0.0001 (****).

**Figure 7 cancers-13-02957-f007:**
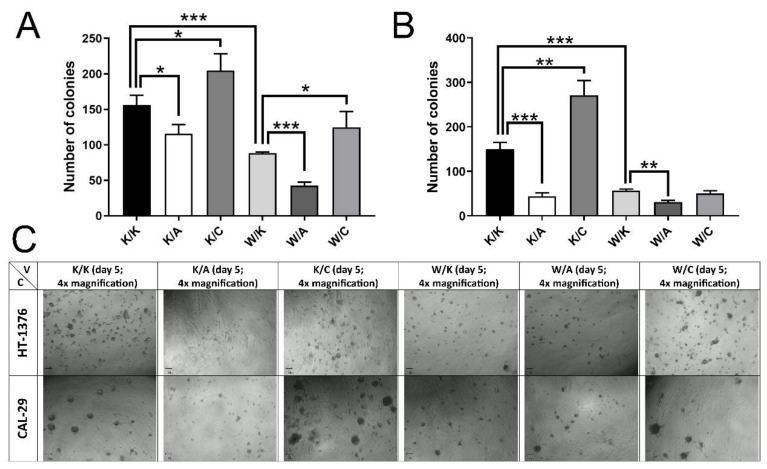
Assessment of changes in the number and size of colonies. (**A**) Colony formation in HT-1376. (**B**) Colony formation in CAL-29. (**C**) Spatial growth in Geltrex. V—Variant. C—Cell line. *p* < 0.05 (*), *p* < 0.01 (**), *p* < 0.001 (***).

**Figure 8 cancers-13-02957-f008:**
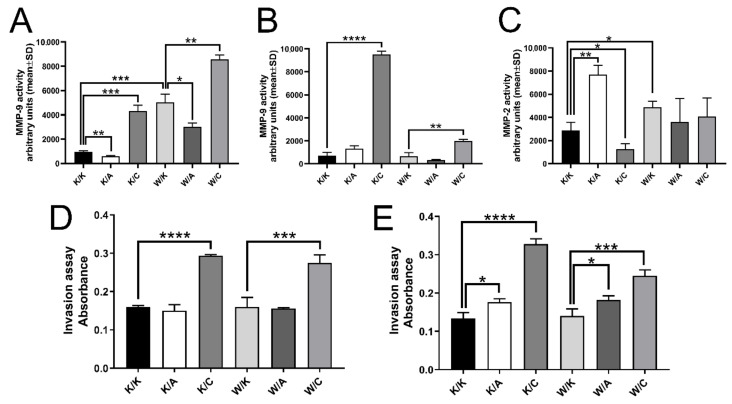
Disparities in the invasiveness potential and MMP-2/9 regulation. (**A**) MMP-9 activity in HT-1376. (**B**) MMP-9 activity in CAL-29. (**C**) MMP-2 activity in CAL-29. (**D**) Invasiveness of HT-1376. (**E**) Invasiveness of CAL-29. *p* < 0.05 (*), *p* < 0.01 (**), *p* < 0.001 (***), *p* < 0.0001 (****).

**Figure 9 cancers-13-02957-f009:**
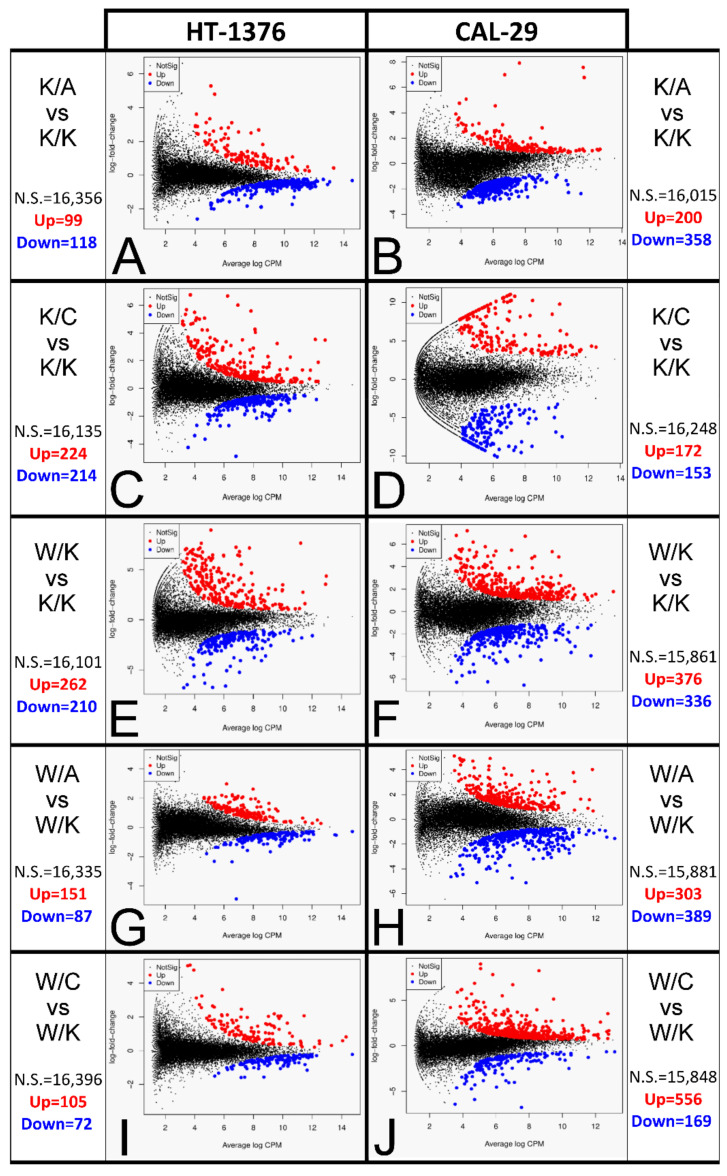
Average genes abundance with log-fold change. (**A**,**B**) K/A vs. K/K. (**C**,**D**) K/C vs. K/K. (**E**,**F**) W/K vs. K/K. (**G**,**H**) W/A vs. W/K. (**I**,**J**) W/C vs. W/K. The comparisons are shown in pairs, i.e., A–B, C–D, E–F, etc.; the paired graphs present data of HT-1376 and CAL-29 cells, respectively.

**Figure 10 cancers-13-02957-f010:**
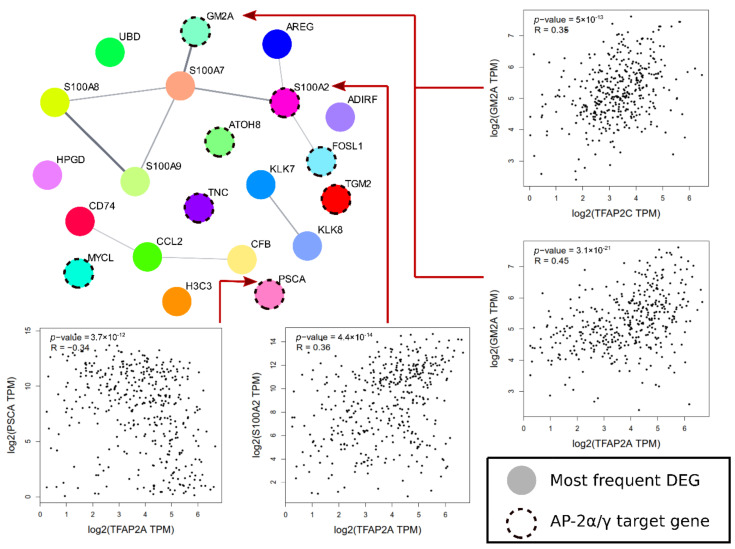
Protein–protein interactions of the most frequent DEGs with a correlation analysis of selected AP-2 targets.

**Figure 11 cancers-13-02957-f011:**
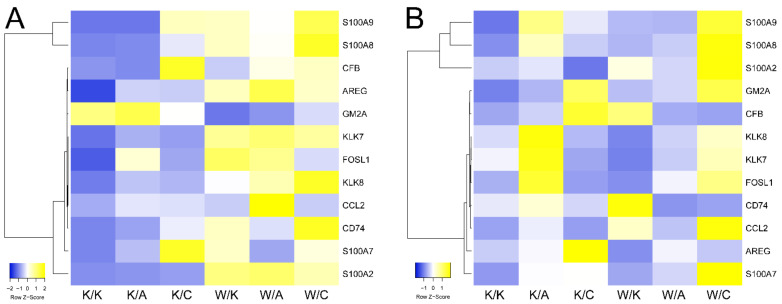
Expression of the most frequent and interacting DEGs across cellular variants. (**A**) HT-1376 variants. (**B**) CAL-29 variants.

**Table 1 cancers-13-02957-t001:** Significance of the most frequent and interacting DEGs regarding bladder cancer.

Gene	Significance in BLCA	Reference
*AREG*	Reduces survival of bladder cancer patients when overexpressed	[29,30]
*CCL2*	Impairs BLCA growth via T-cell recruitment	[31]
*CD74*	Increases invasion, angiogenesis and proliferation of urothelial carcinoma	[32]
*CFB*	None or not yet investigated in BLCA but is strongly expressed in many cancers and its plasma level is useful for prognosis of resectable pancreatic cancers prior to operation	[33]
*FOSL1*	Controls BLCA motility through *AXL* upregulation and is regulated by STAT3, which activates stromal invasion	[34]
*GM2A*	None or not yet investigated, but its overexpression increases the migration of breast cancer cells; supposed prognostic and diagnostic biomarker of lung cancer	[35,36]
*KLK7*	None or not yet investigated, but it might serve as a biomarker of renal papillary carcinoma, which is implicated in colon, thyroid and breast carcinomas; many other kallikreins have an impact on BLCA, e.g., *KLK2/4/5/6/8/9/13*	[37]
*KLK8*	Overexpression decreases the overall survival of BLCA patients	[37]
*S100A2*	Its promoter is methylated more frequently in BLCA patients than healthy individuals; detectable in urine	[38]
*S100A7*	Overexpressed in BLCA with a potential role in tumor progression	[39]
*S100A8*	Differentiates low and high grade BLCA; increases with grading	[40]
*S100A9*	Forms heterodimeric complex (calprotectin) with S100A8; the calprotectin level in urine might serve in BLCA diagnosis and staging as a marker of invasion	[41]

## Data Availability

The dataset(s) supporting the conclusions of this article is (are) included within the article (and its additional file(s)).

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
