# Peer review of "WWOX Loses the Ability to Regulate Oncogenic AP-2γ and Synergizes with Tumor Suppressor AP-2α in High-Grade Bladder Cancer"

_cancers, 2021, doi:10.3390/cancers13122957_

Round 1

Reviewer 1 Report

This study assess the cellular functions of WWOX and AP 2 alpha and AP 2 gamma in two bladder cancer cell lines.   

Authors formerly performed the same experiments in another bladder cancer cell line (RT112). The repetition of the work on two additional cell lines in the present work seems to be less plausible.

The argumentation that these cell lines are different in their grading only weakly justify separate analysis. Did the authors checked whether their cell lines carried the loss of 16q24 locus?

The study is presented in a diffuse way, a more focused discussion would help a better understanding. Six pages of discussion is far too long. The whole manuscript would benefit from shorter but more focused interpretation. 6 pages of discussion is far too long.

A through language edition would be necessary as the text is hard to follow in its present form.

The study contains several formal inconsistencies, only some examples:

  • abbreviations like BLCA should be explained when firs used
  • reference 6: Journal name is missing
  • line 459: refers to section 2.8 which is missing, there is only section 2.8.3 (but no 2.8.2 and 2.8.1)

Own ex vivo data (IHC or gene expression) of the addressed genes would strengthen the results.

Figure 8: Quality data is rather belong to supplementary data and should not included in the main text.

Lines 40-41: 45% loss in BCA based on the estimation of 26 muscle invasive BCa cases, however to my knowledge loss of the WWOX gene was not described in such a high rate in the TCGA study which included WES data of ~400 bladder cancer data. Please check in the publicly available TCGA dataset for bladder cancer.

Author Response

Dear Reviewer 1,

Please see the attachment to download your Response Letter. Thank you for handling peer-review.

With kind regards and on behalf of all Authors,

Damian Kołat

Reviewer 2 Report

Kolat et al. Have studied the effects of WWOX, AP-2alphaA and AP-2gamma overexpression in the two urinary bladder cancer cell lines derived from high grade (grade 3 and 4) tumors i.e. HT-1376 and CAL-29 cells. The study was well planned and carried out and the results are coherent and supportive for the conclusions presented by the authors. The manuscript is well written and could be considered for publication after minor revision and minor spell and language style check. There are some points that should be considered before the editor’s final decision.

  1. Authors should try to shorten the Discussion chapter. The results presented in the study are mostly cohesive and for some methods their interpretation and discussion can be carried out in a single paragraph. The recapitulation of the results could be reduced without affecting the quality of the manuscript. Consider reduction of two first paragraph to the minimum (lines 490-509) and/or transferring them to the introduction chapter.
  2. In a recent study Kaluzinska et al (https://doi.org/10.1186/s12894-021-00806-7) have investigated effects of WWOX overexpression and silencing in the CAL-29 cell lines and it shall be cited and discussed in the revised version of the MS.
  3. Consider to transfer Figure 8 to the supplementary files.

Minor remarks

  1. Values and units shall be separated by space (except for percentage or degrees which are not units).
  2. Line 162 – “Three” from “3D” is added to the chapter number (2.8.3.)
  3. Figures have wrong numeration.
  4. Figures 3, 4 and 5 – it would be easier to follow the Results if type of an assay was given together with the graphs (e.g. next to the Y axis, similarly to the Fig. 7). Similarly, in the Figure 9 the names of experimental groups in comparison (transduction type) shall be given next to the graphs.
  5. Line 809-810: Reference 8 – name of Journal is missing.

Author Response

Dear Reviewer 2,

Please see the attachment to download your Response Letter. Thank you for handling peer-review.

With kind regards and on behalf of all Authors,

Damian Kołat

Reviewer 3 Report

cancers-1233239

General Comments

This manuscript describes the significance of WWOX, AP2α and AP2γ expression in bladder cancer cell lines. Because of the redundant and unfamiliar descriptions, the manuscript is hard to read. In addition, the main purpose of this study is unremarkable. The reviewer feels that the majority of the simply were derived from the previous study (Ref. 8). Please highlight the novelty and significance of this study to cut the meaningless text.

Major Concerns

  1. The significance of grade of bladder cancer is unclear. This study includes only two cell bladder cancer lines, HT-1376 and CAL-29. Are those cell lines truly representative grade 3 and 4 bladder cancer, respectively? Are the results derived from the present study associated with difference of the grade? The reviewer considers that the difference may be simply inter-tumoral heterogeneity and/or diversity, and it is NOT directly related to the grade of bladder cancer. Please also define grade of bladder cancer in manuscript.

  1. Figure 11. Events of each transduction are unclear. Please explain them in figure legend or amend the figure.

  1. Please clarify the provider or reference of all the materials and methods used in the present study.

  1. Generally, the manuscript, especially data descriptions and discussion, is redundant despite the in vitro study of only two cell lines. Please revise the text concisely.

Minor Points

  1. Abstract. Please expand BLCA for the first-time use.
  2. Bibliographic information of Ref. 8 is incomplete. Please confirm that all the references are correct.
  3. Page 3. Figure should appear in numerical order.
  4. Page 4. “2.8.3” seems to be “2.8”.
  5. Pages 5-6. 2.13. DGEList(), topTags() and plotMD() are unclear.
  6. Detailed p-values seem to not be important.
  7. Figure 8 may be appropriate for supplementary materials, because it has a little impact on the main results of the present study.

Author Response

Dear Reviewer 3,

Please see the attachment to download your Response Letter. Thank you for handling peer-review.

With kind regards and on behalf of all Authors,

Damian Kołat

Reviewer 4 Report

The authors analyzed the effect of WWOX, AP2-alpha, and AP2-gamma on cell proliferation, apoptosis, adhesion, and self renewal in grade 3 and 4 bladder cancer cell lines. They recently published a similar paper using a grade 2 bladder cancer cell line. The effects of ectopic WWOX/AP2a/AP2g on proliferation, apoptosis, adhesion, self renewal were generally stronger in grade 3/4 cell lines than in the grade2 cell line. However, some effects as for example enhanced apoptosis upon ectopic WWOX expression or decreased colony formation upon ectopic AP2a also occured in the grade 2 bladder cancer cell line. Moreover, the effects of ectopic WWOX/AP2a/AP2g on proliferation, apoptosis, adhesion, self renewal are not always consistent between the analyzed grade 3 and 4 cell lines. Therefore, by analyzing only one cell line for each grade it can not be generalized that WWOX/AP2a/AP2g have specific functions in grade 2 and grade 3/4 bladder cancers. Maybe the effects are just cell line dependent independent of grading.

- Figure 1: The verification of ectopic WWOX/AP2a/AP2g expression is evident, but the quality of western blots could be improved.

- section 3.10.: In the majority of assays AP2a and AP2g have opposite functions. This should be considered when DEGs that appear in several comparisons (i.e. K/A vs K/K ....) are identified.

- Table 1: for each gene it should be indicated in which cell variants is it up- or down-regulated.

- Figure 10: Do the correlation plots represent expression in tumors?

- Discussion is too long. The results don't have to be repeated.

Author Response

Dear Reviewer 4,

Please see the attachment to download your Response Letter. Thank you for handling peer-review.

With kind regards and on behalf of all Authors,

Damian Kołat

Round 2

Reviewer 1 Report

I really appreciate the hard work of the authors that was put in the revision of this manuscript.

However, the general problem that the authors concluded differences by assessing one cell line for each tumor grade is not appropriate. The observations that they describe may be well be simply cell line dependent and less grade dependent.

Author Response

Dear Reviewer 1,

Thank you so much for appreciating our work. We sincerely apologize for not covering this issue, but in Authors opinion, such advice did not reach us in your previous peer-review round. Nevertheless, we will be happy to provide justification.

  • Although it is not possible to entirely exclude the impact of other factors, we carefully examined whether these cell lines are both comparable between each other (to eliminate potential inter-heterogeneity and/or diversity) and at the same time diverse in terms of characteristic we aimed to investigate.
  • As we mentioned previously, these two cell lines possess D16S539 short tandem repeat (please see Cellosaurus resource, lines IDs: CVCL_1292, CVCL_1808). Both HT-1376 and CAL-29 are representative cell lines for respectively grade 3 and 4 BLCA, and are used as experimental models (please see PMID 17970057).
  • Moreover, they do possess consistent molecular profile in terms of TP53, HRAS, NRAS, KRAS yet they vary in terms of PI3KCA – it is mutated in CAL-29 but not in HT-1376. However, this change in PI3KCA can be explained by the fact, that mutation of this gene is present in very high grade of various tumors, hence such slight differences are inevitable when investigating various grades.
    • Catasus et al. wrote that: “Previous studies in various epithelial cancers have found that PIK3CA mutations are mainly associated with high-grade and invasive tumors. For instance, in colorectal carcinomas, their frequency was significantly greater in high-grade and advanced carcinomas (32%) than in low-grade tumors (3%). Similarly, a higher rate of PIK3CA mutations was found in invasive lobular and ductal breast carcinomas than in other histologic types associated with more favorable prognosis.” (please see DOI:10.1038/modpathol.3800992)
  • Additionally, both models used in this study possess staging T2, please see PMID 17970057.
  • Next, the impact of grade is evident on all three individuals i.e. WWOX, AP-2α and AP-2γ (we described it in your previous response letter and the current manuscript version is now updated to underline this fact).
  • Furthermore, in our opinion the differences between variants in both cell lines are mainly consistent (if significant), while few discrepancies were explained through some sort of shift in superiority or partnership (we devoted whole paragraph in “Discussion” to explain our presumptions). In a global view and using single sentence, we observed that WWOX might gradually losing control over AP-2γ which manifests its oncogenic character; at the same time, the nature of WWOX or AP-2α and their collaboration has the anti-cancer net effect.
  • Undoubtedly, decision to extend methodology with more than one cell line per grade would strengthen conclusions. However, due to abundance of cellular variants and comparisons between them in current state, we felt that having multiplication of variants for WWOX and TFAP2A/C would be so complicated that it will be difficult to explain all the dependencies. Such complexity is already visible through the results and the final inference based on them. Many times, the use of more than one cell line in the experiment strengthens the conclusions in the case of one variable (e.g. gene/protein) and we agree such maneuver is then recommended – however in our case, we conclude about the function of three variables (WWOX, AP-2α and AP-2γ) at once and additionally their collaboration.

Hopefully the explanation and justification of our decision will be sufficient to meet the requirements. Thank you for handling peer-review.

With kind regards and on behalf of all Authors,
Damian Kołat

Reviewer 3 Report

The authors properly answer the comments of the reviewers, and the revised manuscript have been considerably improved for publication. 

Author Response

Dear Reviewer 3,

Thank you so much for opinion!

With kind regards and on behalf of all Authors,
Damian Kołat

Reviewer 4 Report

Q1/Figure 1: ok

Q3/Supp. File 3: It is not clear whether a gene is up- or down-regulated DEG in a comparison (i.e. K/A vs K/K). It would be better to make a heatmap with fold changes or log2 fold changes for each comparison and each gene.

Q4/Table 1: in the legend it should be better explained what the columns 4 and 5 show, like in the response letter. If I understand correctly, all genes are upregulated versus control. How about the downregualted genes? It would be better to show an extra figure with a colored heatmap showing normalized expression of all genes from table 1 in all cell variants.

Q5/Figure 10: It should be indicated that the BLCA tumor dataset is from TCGA.

Q6/Discussion: ok

Author Response

Dear Reviewer 4,

Thank you so much for your opinion. We are very pleased that Q1 and Q6 are now clarified.

  • Regarding Q5, we have added information about TCGA, please see line 270. To clarify, GEPIA2 contains only TCGA data in terms of tumors, that is why we did not mention it previously. But in the current form, it is fully informative. Thank you for your advice.
  • For Q3, we updated Supp. File 3 by adding heatmap* to the previous data. You can see the final effect via opening the supplementary file itself.
  • In response to Q4, new figure in the main text (Figure 11, see line 504) is now included and contains colored heatmap* as suggested. We felt that showing similar data on Table 1 and Figure 11 will be considered as repetition, thus we deleted columns 4 and 5 from table. Through that, better explanation (like in response letter) is not needed at the current stage.
    • Just to clarify, the columns 4 and 5 already contained information about downregulated genes but control variant K/K was most of the time found as the last variant in descending list (there were only a few K/K located not in the end of the list, so probably these might have been unspotted).

* For these two questions, heatmaps were generated using Heatmapper. We included new methodology section to describe it properly (see line 272).

We hope that the changes made throughout the entire manuscript will be sufficient to meet the requirements. Thank you for handling peer-review.

With kind regards and on behalf of all Authors,
Damian Kołat